

# Identification of two *CiGAD*s from *Caragana intermedia* and their transcriptional responses to abiotic stresses and exogenous abscisic acid

Jing Ji[1,*], Lingyu Zheng[1,2,*], Jianyun Yue[1], Xiamei Yao[1], Ermei Chang[1], Tiantian Xie[1], Nan Deng[1], Lanzhen Chen[3,4], Yuwen Huang[5], Zeping Jiang[1] and Shengqing Shi[1]

[1] State Key Laboratory of Tree Genetics and Breeding, Chinese Academy of Forestry, Research Institute of Forestry, Beijing, China
[2] Chongqing University of Technology, Chongqing, China
[3] Institute of Apicultural Research, Chinese Academy of Agricultural Sciences, Beijing, China
[4] Risk Assessment Laboratory for Bee Products, Quality and Safety of Ministry of Agriculture, Beijing, China
[5] The High School Affiliated to Renmin University of China, Beijing, China
[*] These authors contributed equally to this work.

Corresponding authors
Zeping Jiang, jiangzp@caf.ac.cn
Shengqing Shi, shi.shengqing@caf.ac.an

## ABSTRACT

**Background**. Glutamate decarboxylase (GAD), as a key enzyme in the $\gamma$-aminobutyric acid (GABA) shunt, catalyzes the decarboxylation of L-glutamate to form GABA. This pathway has attracted much interest because of its roles in carbon and nitrogen metabolism, stress responses, and signaling in higher plants. The aim of this study was to isolate and characterize genes encoding GADs from *Caragana intermedia*, an important nitrogen-fixing leguminous shrub.

**Methods**. Two full-length cDNAs encoding GADs (designated as *CiGAD1* and *CiGAD2*) were isolated and characterized. Multiple alignment and phylogenetic analyses were conducted to evaluate their structures and identities to each other and to homologs in other plants. Tissue expression analyses were conducted to evaluate their transcriptional responses to stress (NaCl, $ZnSO_4$, $CdCl_2$, high/low temperature, and dehydration) and exogenous abscisic acid.

**Results**. The *CiGAD*s contained the conserved PLP domain and calmodulin (CaM)-binding domain in the C-terminal region. The phylogenetic analysis showed that they were more closely related to the *GADs* of soybean, another legume, than to *GADs* of other model plants. According to Southern blotting analysis, *CiGAD1* had one copy and *CiGAD2*-related genes were present as two copies in *C. intermedia*. In the tissue expression analyses, there were much higher transcript levels of *CiGAD2* than *CiGAD1* in bark, suggesting that *CiGAD2* might play a role in secondary growth of woody plants. Several stress treatments (NaCl, $ZnSO_4$, $CdCl_2$, high/low temperature, and dehydration) significantly increased the transcript levels of both *CiGAD*s, except for *CiGAD2* under Cd stress. The *CiGAD1* transcript levels strongly increased in response to Zn stress (74.3-fold increase in roots) and heat stress (218.1-fold increase in leaves). The transcript levels of both *CiGAD*s significantly increased as GABA accumulated during a 24-h salt treatment. Abscisic acid was involved in regulating the expression of these two *CiGAD*s under salt stress.

**Discussion**. This study showed that two *CiGAD*s cloned from *C. intermedia* are closely related to homologs in another legume, soybean. *CiGAD2* expression was much higher than that of *CiGAD1* in bark, indicating that *CiGAD2* might participate in the process of secondary growth in woody plants. Multiple stresses, interestingly, showed that Zn and heat stresses had the strongest effects on *CiGAD1* expression, suggesting that *CiGAD1* plays important roles in the responses to Zn and heat stresses. Additionally, these two genes might be involved in ABA dependent pathway during stress. This result provides important information about the role of *GAD*s in woody plants' responses to environmental stresses.

# INTRODUCTION

$\gamma$-Aminobutyric acid (GABA) is a non-protein amino acid present in animals, plants, and other organisms (*Batushansky et al., 2014*; *Gilliham & Tyerman, 2016*; *Michaeli & Fromm, 2015*; *Shelp et al., 2012a*; *Shelp et al., 2012b*). It functions as a major inhibitory neurotransmitter in the brain tissues of mammals (*Granger et al., 2016*; *Robel & Sontheimer, 2016*). In plants, GABA plays roles in diverse processes including the carbon:nitrogen balance, signaling, regulation of redox status, development, stress responses (*Batushansky et al., 2014*; *Batushansky et al., 2015*; *Michaeli & Fromm, 2015*; *Molina-Rueda et al., 2015*; *Shi et al., 2010*; *Yu et al., 2014*). Recent studies have indicated that GABA is required for proper growth during exposure to abiotic stresses such as salt (*Renault et al., 2013*), drought (*Mekonnen, Flügge & Ludewig, 2016*), low temperature (*Aghdam et al., 2016*), $Zn^{2+}$ (*Daş et al., 2016*), and cadmium (*Sun et al., 2010*). Additionally, exogenous GABA has been shown to regulate the expression of genes such as *BnNrt2* in *Brassica napus* (*Beuve et al., 2004*), *14-3-3* in *Arabidopsis thaliana* (*Lancien & Roberts, 2006*), *SAMDC* (S-adenosylmethionine decarboxylase) in *Cucumis melo* (*Wang et al., 2014*), and to control gene transcription in *Caragana intermedia* (*Shi et al., 2010*) and *A. thaliana* (*Batushansky et al., 2014*). The metabolite GABA has been shown to play roles in a diverse range of cellular processes ranging from neuronal inhibition in animals to pollen-tube development in plants (*Michaeli & Fromm, 2015*). Consequently, many researchers have focused on the production of GABA, and in particular, the role of glutamate decarboxylase (GAD), which is the unique pyridoxal enzyme catalyzing the α-decarboxylation of L-glutamate to form GABA (*Bouché & Fromm, 2004*; *Shelp et al., 2012a*; *Shelp et al., 2012b*).

The pathway of glutamate decarboxylation catalyzed by GAD is widely distributed in organisms (*Liu et al., 2014*), and is considered to be the main pathway of GABA production in plants (*Shelp et al., 2012a*). Genes encoding GAD enzymes have been successfully identified in many herbaceous plants such as rice (*Akama et al., 2001*), *Arabidopsis* (*Bouché et al., 2004*), maize (*Zhuang et al., 2010*), and *Panax ginseng* (*Lee et al., 2010*) and in some woody plants, such as pine (*Molina-Rueda et al., 2010*), apple (*Trobacher et al., 2013*), citrus (*Liu et al., 2014*), and tea (*Mei et al., 2016*). Bioinformatics analyses and experimental

characterizations have shown that plant GADs contain a calmodulin (CaM)-binding domain that is generally responsible for the cytosolic decarboxylation of glutamate to GABA (*Shelp et al., 2012a*), although one functional *OsGAD2* without a $Ca^{2+}$/calmodulin domain has been isolated from rice (*Akama et al., 2001*). Several studies have demonstrated spatiotemporal differences in the expressions of *GAD* gene family member, although most *GAD* families studied to date have fewer than nine members (*Shelp et al., 2012a*). For example, *AtGAD1* is mainly expressed in *Arabidopsis* roots (*Miyashita & Good, 2008*), while *ZmGAD1* is expressed in the leaves, stems, and roots of maize (*Zhuang et al., 2010*). In rice, *OsGAD1* is mainly expressed in seeds but *OsGAD2* is mainly expressed in roots (*Akama et al., 2001*). In citrus, *CsGAD1* is predominantly expressed in flowers but *CsGAD2* is predominantly expressed in fruit (*Liu et al., 2014*). Interestingly *PpGAD* expression was shown to be correlated with vascular differentiation in pine seedlings (*Molina-Rueda et al., 2010*). Additionally, environmental stimuli were shown to induce the expression of *ZmGAD1* in maize (*Zhuang et al., 2010*) and *PgGAD* in *P. ginseng* (*Lee et al., 2010*). These studies indicate that plant *GAD*s might play important roles in plant development and stress responses. However, few studies have focused on the expression of *GAD*s in woody plants under environmental stresses.

Our previous studies showed that salt stress induced significant expression of EST sequences (Supplemental Information 1) homologous to plant *GAD* genes in the desert legume shrub, *C. intermedia.* We also showed that exogenous GABA can enhance the expression of genes, including those related to hormones and reactive oxygen species (ROS), which involved the stress responses of this shrub (*Shi et al., 2010*). Therefore, in the present study, we cloned two full-length cDNAs, *CiGAD1* and *CiGAD2*, from *C. intermedia* based on known EST sequences using the rapid amplification of cDNA ends (RACE) technique. We analyzed changes in their transcript levels in response to different stress factors and exogenous abscisic acid (ABA). These results allowed us to explore the relationship between *GAD* expression and GABA accumulation, which will increase our understanding of the roles of GABA in stress adaptation in plants.

## MATERIALS AND METHODS

### Plant materials and treatments

Seeds of *C. intermedia* were collected from a desert habitat in Inner Mongolia, China. Seedlings were cultivated as described by *Shi et al. (2010).* Briefly, seeds were washed and then germinated in sand. Three-day-old seedlings (radical length, 30–40 mm) were selected and cultivated with water in pots in a growth chamber under a 14-h light/10-h dark photoperiod and a day/night temperature of 23/18 °C. The seedlings were grown for 2–3 weeks, and the water was refreshed every 2 days. Subsequently, the seedlings were subjected to the following stress treatments: (1) NaCl (200 mM), $ZnSO_4$ (100 μM), $CdCl_2$ (500 μM), high temperature (40 °C), low temperature (4 °C), or dehydration (for 3 h); (2) NaCl (200 mM) for different durations (0, 3, 6, 12, and 24 h); and (3) exogenous ABA (0, 0.5, 1, 10, 100, and 200 μM) combined with the 200 mM NaCl treatment for 0, 3, and 24 h. Roots and leaves were harvested, frozen immediately in liquid nitrogen, and then stored

at −80 °C. Also, different tissues including roots, stems, leaves, seeds, testae, and bark were collected from mature individuals growing at the Chinese Academy of Forestry, Beijing, frozen immediately in liquid nitrogen, and then stored at −80 °C. Every experimental treatment had three biological replicates.

## Cloning, sequencing, and bioinformatic analysis

Total RNA was extracted from pooled roots and leaves with Trizol reagent (Invitrogen, Carlsbad, CA, USA), and complementary DNA (cDNA) was generated with the SMARTer™ RACE cDNA amplification kit (Clontech, Palo Alto, CA, USA) according to the manufacturer's instructions. Then, 5′ and 3′ RACE were performed using nested gene-specific primers (Table S1). The obtained amplification products were cloned into the pMD19-D vector (Takara, Dalian, China) and sequenced. Then, the two full-length sequences were assembled by DNAMAN 6.0. *CiGAD1* and *CiGAD2* have been deposited in GenBank under the accession numbers KU586714 and KU586715, respectively.

The two full-length cDNA sequences of *CiGAD1* and *CiGAD2* were analyzed with ORF Finder (https://www.ncbi.nlm.nih.gov/orffinder/), and the isoelectric point (pI) and molecular weight (MW) of the predicted protein were estimated using ExPASy (http://web.expasy.org/compute_pi/). Sequences for the protein homology analysis were obtained from phytozome (https://phytozome.jgi.doe.gov/pz/portal.html), and aligned using DNAMAN 8.0. The phylogenetic tree was constructed using MEGA 6.06 software by the maximum-likelihood algorithm with 1,000 bootstrap replicates.

## Southern blot analysis

Genomic DNA was isolated from leaves of *C. intermedia* using CTAB method (*Porebski, Bailey & Baum, 1997*), and 20 μg DNAs were digested with BamH I and EcoR V, respectively. The obtaining fragments were separated by electrophoresis through a 1.0% agarose, then the DIG-labeled DNA probes of *CiGAD1* and *CiGAD2* were hybridized followed the protocol of DIG-High Prime DNA Labeling and Detection Starter Kit I (Roche Applied Science, Mannheim, Germany). Briefly, the DIG-labeled DNA probe was prehybridized at 60 °C for 2 h and then hybridized 37 °C for 12 h. Following hybridization, the blot was washed twice in 2× SSC containing 0.1% SDS at 25 °C for 5 min and was washed twice in 0.1× SSC containing 0.1% SDS at 50 °C for 15 min. The immunodetection of the DIG-labeled probe was performed with 1:10,000 anti DIG-AP. The blot was exposed to X-ray film (AGFA, Germany). DNA probes were amplified from genomic DNA by using the primers designed in the sequenced DNA fragments based on the cDNA sequences of *CiGAD1* and *CiGAD2*, while the probe for *CiGAD1* was mainly in the region from 1,076–1,370 bp in the full-text cDNA, and the probe for *CiGAD2* was mainly in the region from 639–1,031 bp in the full-text cDNA. Primers are listed in Table S1 and probe sequences are listed in Supplemental Information 3.

## Assay of GAD activity

About 0.1 g sample (root, stem, leaf, seed, or bark) was ground into a powder in liquid nitrogen and transferred to a 2-mL centrifuge tube. Then, 1 mL pre-cooled extraction buffer (150 mM potassium phosphate, 5 mM EDTA, 1 mM magnesium chloride, 0.5% (w/v)

polyvinylpyrrolidone, 3 mM 2-mercaptoethanol, 0.2 mM PLP (pyridoxal 5′-phosphate), 10% (v/v) glycerol, 1 mM phenylmethylsulfonyl fluoride, pH 5.8) was added, and the homogenate was mixed well. After 15 min incubation on ice, the homogenate was centrifuged at 10,000 g for 15 min at 4 °C. The supernatant was collected into a 2-mL volumetric flask and the volume was adjusted to 1 mL. This was used as the crude enzyme solution (CES; *Liu et al., 2014*).

To measure GAD activity, the assay mixture (1 mL) contained 0.4 mL 0.1% sodium glutamate, 0.2 mL 0.25 mM PLP, and 0.4 mL CES. After mixing well, the reaction mixture was incubated at 40 °C for 5 h and then boiled for 5 min to end the reaction. Sodium glutamate was replaced by 0.4 mL distilled water in the blank control. After cooling, the absorbance of the mixture was measured at 640 nm using a Multi-Mode Detection Platform (SpectraMax Paradigm; Molecular Devices Co., Sunnyvale, CA, USA). A GABA standard curve was constructed and used to calculate the GABA content in the samples based on absorption value, as described by *Yang, Hui & Gu (2016)*. And protein contents were tested by Bradford method (*Kruger, 1994*) with bovine serum albumin as standard. One unit (U) was defined as the amount ($\mu$mol) of GABA produced by the action of one mg protein per hour. All determinations were representative of three biological experiments and four technical replicates.

## Determination of GABA

The plant tissues were ground to a fine powder in liquid nitrogen, and about 0.1 g of the frozen homogenate was extracted using methanol and lanthanum chloride, and then centrifuged at 13,000g for 5 min, and 0.8 ml of the supernatant was transferred to a new Eppendorf tube. The 160 $\mu$l of 1 M KOH was added, and centrifugation as before. The 100 $\mu$l resulting supernatant was used in the spectrophotometric GABA determination (*Zhang & Bown, 1997*). The assay mixture (300 $\mu$l) also contained 75 mM potassium pyrophosphate (pH 8.6), 3.3 mM 2-mercaptoethanol, 1.25 mM $\beta$-NADP$^+$, 5 mM 2-ketoglutarate and 0.02 units of GABase (Sigma). The increase of OD340 nm was recorded using 96-well microplate reader. The amount of GABA was calculated according to external calibration curve of GABA (*Renault et al., 2010*). Values shown were representative of three biological experiments and four technical replicates.

## qRT-PCR

Total RNAs were extracted from a pool of roots and leaves with Trizol reagent (Invitrogen), and then cDNAs were synthesized with 0.5 $\mu$g RNAs using the Prime Script$^{TM}$ RT reagent kit (Perfect Real Time; Takara, Dalian, China) according to the manufacturer's instructions. Specific RT-PCR primers (Table S2) were designed to have melting temperatures of 60 °C and amplicon lengths of 150–200 bp using Primer3 software (http://primer3.ut.ee/). Real-time qRT-PCR was performed in quadruplicate using the SYBR Premix Ex Taq$^{TM}$ II kit (Takara) on a Roche lightCycler 480 (Roche Applied Sciences, Penzberg, Germany) according to the manufacturer's instructions. The $2^{-\Delta Ct}$ method was adopted to analyze the qRT-PCR result, and $\Delta Ct = Ct_{target}$-$Ct_{EF1\alpha}$. Values shown were representative of three biological experiments.

## Statistical analysis

Data were evaluated by Duncan's multiple test in SPSS v.19. Differences were considered significant at $P < 0.05$.

# RESULTS

## Isolation of *CiGAD*s and bioinformatics analyses

Two full-length cDNAs of glutamate decarboxylase (*GAD*) genes were isolated from *C. intermedia* by 5′- and 3′-RACE, as shown in Fig. S1, and designated as *CiGAD1* (GenBank ID: KU586714) and *CiGAD2* (GenBank ID: KU586715). Bioinformatics analyses showed that *CiGAD1* had a 1922-bp full-length mRNA sequence containing a 1521-bp open reading frame (ORF), a 107-bp 5′ puntranslated region (UTR), and a 294-bp 3′-UTR; and that *CiGAD2* had a 1797-bp full-length mRNA sequence containing a 1797-bp ORF, a 60-bp 5′-UTR, and a 237-bp 3′-UTR. The alignment showed that the nucleotide identity between *CiGAD1* and *CiGAD2* was 69.0%. Theoretically, *CiGAD1* encodes a 57.3-kDa peptide consisting of 506 amino acids with the theoretical isoelectric point (pI) of 5.5, and *CiGAD2* encodes a 56.1-kDa peptide consisting of 499 amino acids with the theoretical isoelectric point (pI) of 5.6.

The deduced amino acid sequences of the two CiGADs were aligned with GAD family members from three typical model plants; *Arabidopsis*, *Glycine*, and *Populus* (Fig. S2). The amino acid identity between CiGAD1 and CiGAD2 was 71.4%. CiGAD1 shared 88.5%, 79.9% and 79.4% amino acid identity with *Glycine* GAD5, *Arabidopsis* GAD1, and *Populus* GAD1, respectively. CsGAD2 shared 83.1%, 75.0%, and 72.1% amino acid identity with *Glycine* GAD2, *Populus* GAD5, and *Arabidopsis* GAD5, respectively. Both CiGAD proteins contained two conserved domains: a PLP-binding domain in the middle region and a CaM-binding domain at the carboxyl terminus (Fig. 1). The ML tree showed that CiGAD1 was located close to GmGAD5, GmGAD1, and GmGAD3; while CiGAD2 was in a completely different clade, close to GmGAD2, GmGAD4, PtGAD5, and AtGAD5 (Fig. 2). These findings indicated that CiGAD proteins from *C. intermedia*, a salt- and drought-resistant desert legume shrub, had higher homologies with GAD homologs in soybean, another well-known legume.

We also performed a southern blot analysis to estimate the copy number of two *GAD* genes in *C. intermedia*. The result showed one band in BamH I or EcoR I restriction the digest lane of *CiGAD1* (Fig. 3A) and two bands in the BamH I or EcoRI restriction digest lane of *CiGAD2* (Fig. 3B), respectively. This confirmed that *CiGAD1* had one copy, and *CiGAD2*-related genes were present as two copies in the genome of *C. intermedia*.

## Tissue-specific expression of two *CiGAD*s

The expression of two *CiGAD*s showed different patterns in various issues including roots, stems, leaves, seeds, testae, and bark from mature individuals (Fig. 4A). The transcript levels of *CiGAD1* were lower than those of *CiGAD2* in roots, seeds, testae, and bark, but the opposite pattern was observed in other tissues. The two genes showed contrasting trends in mRNA abundance in most tissues. The highest transcript level of *CiGAD1* was in testae, followed by leaves and stems, while it was almost undetectable in roots and bark. In

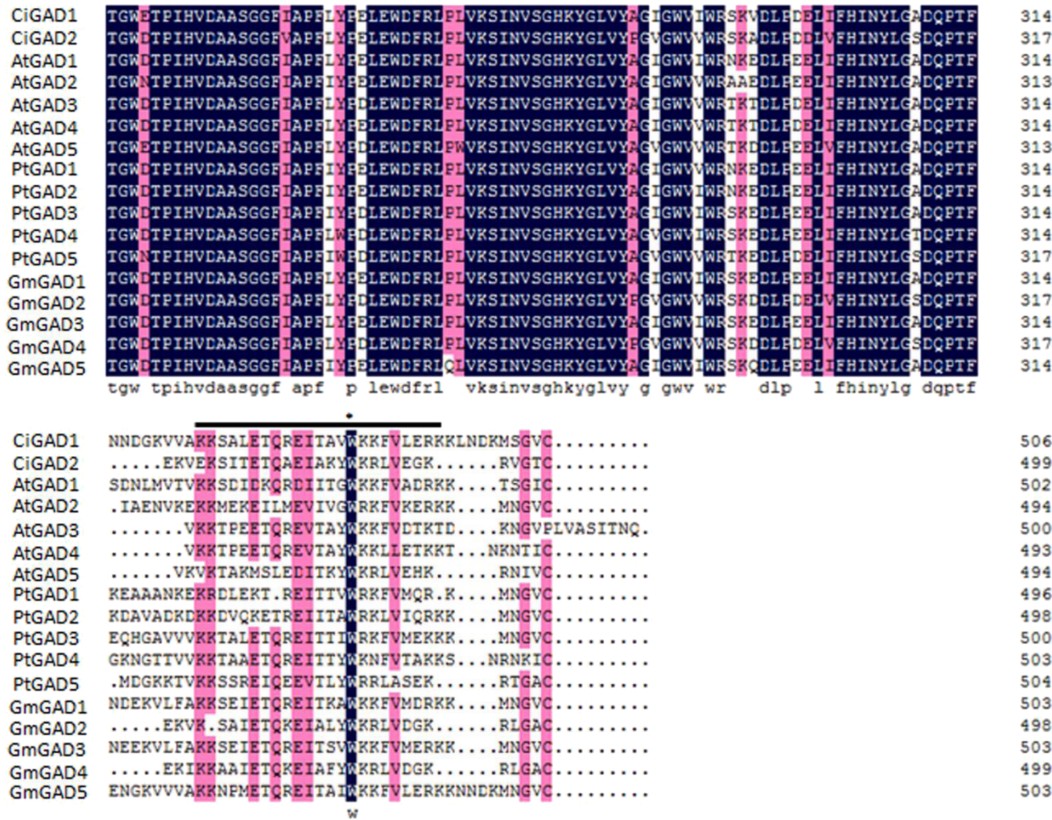

**Figure 1** Alignment of the deduced amino acid sequence of two CiGADs in *Caragana intermedia* with the deduced amino acid residues of GADs in *Arabidopsis thaliana, Populus trichocarpa, Glycine max.* Accession number: AtGAD1 (AT5G17330), AtGAD2 (AT1G65960), AtGAD3 (AT2G02000), AtGAD4 (AT2G02010) AtGAD5 (AT3G17760), GmGAD1 (Glyma02g40840), GmGAD2 (Glyma09g29900), Gm-GAD3 (Glyma14g39170), GmGAD4 (Glyma16g34450), GmGAD5 (Glyma18g04940), PtGAD1 (Potri.T059200), PtGAD2 (Potri.004G075200), PtGAD3 (Potri.004G075300), PtGAD4 (Potri.010G100500), Pt-GAD4.1 (Potri.008G141100), and PtGAD5 (Potri.012G039000). The abbreviation of gene names are as follows: At, *Arabidopsis thaliana*; Pt, *Populus trichocarpa*; Gm, *Glycine max.* Identical and similar amino acid residues were shown in black or pink, respectively. The pyridoxal-5-phosphate binding domain of the GADs was underlined with a thin line, and the CaM binding domain was underlined with a thick line; Trp (W), the important site for *in vitro* binding to CaM, was indicated by an asterisk.

contrast, *CiGAD2* showed the highest transcript level in the bark (37.9-fold that of *CiGAD1* in the bark), followed by testae and roots (1.9 and 21.3 fold higher, respectively, than that of *CiGAD1*). The GAD activity was highest in testaes (9.9 U), followed by seeds (3.2 U). But its activity was lower, and similar, among the other tested tissues (1.1–1.9 U) (Fig. 4B).

## Stress-specific expression of two *CiGADs*

To investigate whether the two *CiGADs* were expressed differently in response to various stress conditions, total RNA was extracted from roots and leaves of young seedlings under several different stress treatments. Both *CiGADs* showed dramatic increases in their transcript levels in roots and leaves in response to stress (Figs. 5A, 5B ). Under Zn stress, the transcript level of *CiGAD1* increased by 74.3-fold and that of *CiGAD2* increased by 6.3-fold, compared with their respective levels in control roots. *CiGAD1* showed 3.4–11.3

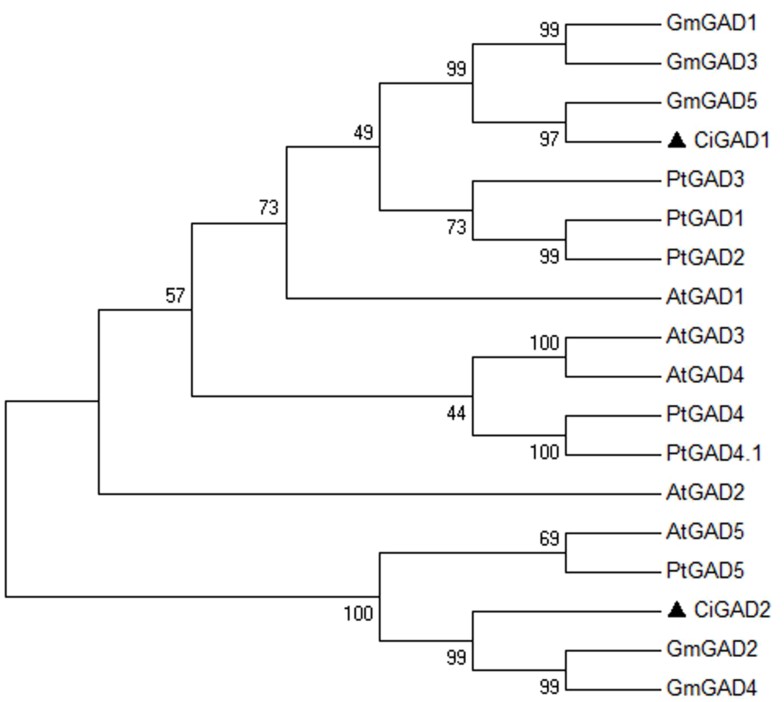

**Figure 2** **Phylogenetic analysis of GAD sequences from Caragana intermedia and other three model plants *Arabidopsis thaliana, Populus trichocarpa, Glycine max*.** The consensus tree was obtained by the Maximum-Likelihood method in MEGA 6.06. A bootstrap analysis of 1,000 replicates was performed. Accession numbers were as shown in the legend of Fig. 1. The candidate GADs were marked with black triangle.

fold increases in its transcript levels in roots under dehydration, Cd, Na, and low/high temperature stresses. *CiGAD2* transcript levels showed 1.0–5.5 fold increases in roots under all of the stresses except Cd stress (Fig. 5A). In leaves, the transcript levels of *CiGAD1* and *CiGAD2* strongly increased in response to high temperature by 218.1-fold and 114.2-fold, respectively, compared with their respective levels in the control leaves (Fig. 5B). The transcript levels of *CiGAD1* and *CiGAD2* increased by 20.6-fold and 36.0-fold, respectively, under NaCl stress, and by 19.7-fold and 21.8-fold, respectively, under low temperature stress. Under Zn, Cd, and dehydration stress, *CiGAD1* expression increased by 4.4–8.5 fold, and *CiGAD2* expression increased by 1.5–5.2 fold. These results indicated that the transcript levels of *CiGAD1* and *CiGAD2* increased to varying degrees in response to various stresses.

To analyze the expression of the two *CiGAD*s under stress in more detail, seedlings were treated with NaCl for different times, as shown in Fig. 6. The transcript levels of *CiGAD1* and *CiGAD2* significantly increased as the duration of the NaCl treatment extended, although there were some fluctuations in their transcript levels in leaves at 12 h and in roots at 6 h (Figs. 6A and 6B). In roots, the transcript levels of *CiGAD1* and *CiGAD2* had increased by 20.8-fold and 16.4-fold, respectively, after 6 h of salt stress, and by 41.2-fold and 12.0-fold, respectively, after 24 h of salt stress, compared with their respective levels at 0 h (Fig. 6A). However, in the leaves, *CiGAD1* and *CiGAD2* transcript levels remained relatively stable except at 6 h of salt stress, when they were increased by 20.6–23.7 fold, and

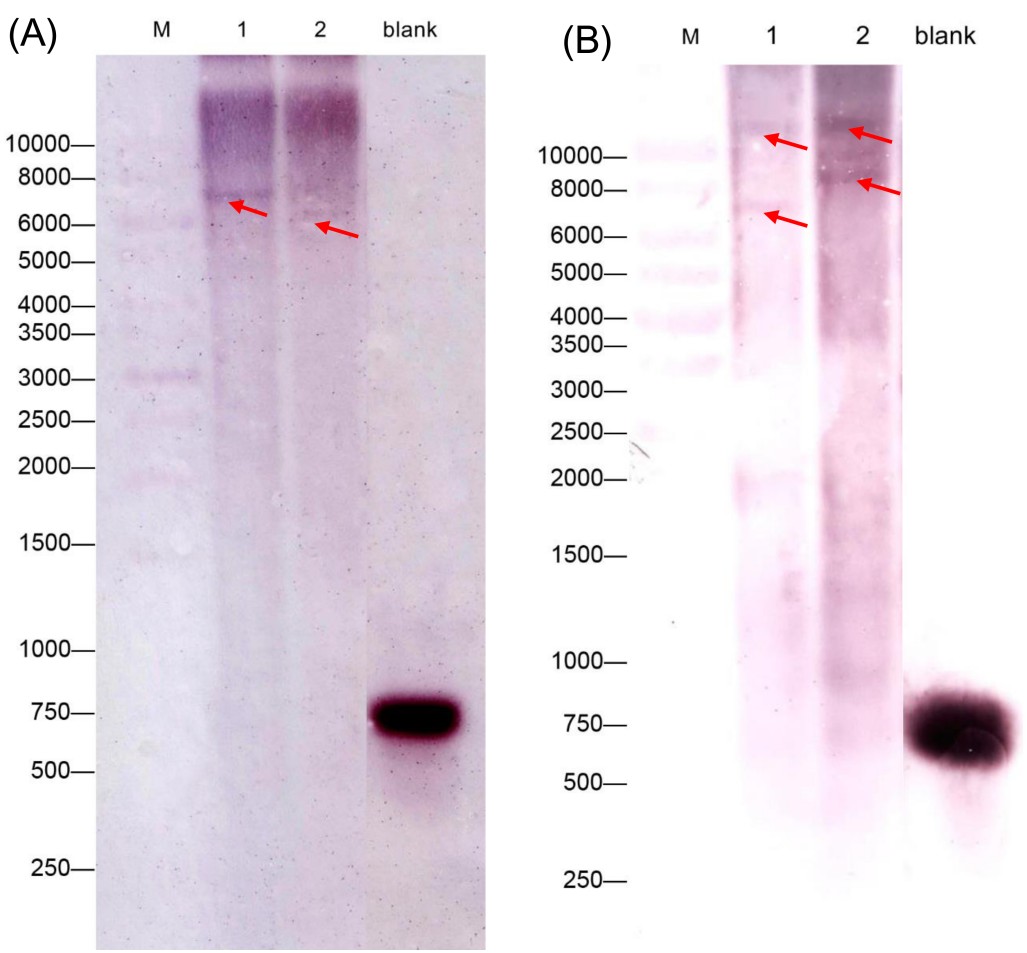

**Figure 3** **Southern blotting analysis of *CiGAD1* (A) and *CiGAD2* (B).** Lane 1: BamH I; Lane 2: EcoR V. Red arrows were targeted bands; M: 10,000 bp DNA marker; blank: DNA probe.

11.5–13.8 fold, respectively, compared with their respective levels at 0 h (Fig. 6B). Further analyses showed that *CiGAD1* was more strongly induced than was *CiGAD2* as the NaCl treatment extended. Corresponding to the expression of *CiGADs*, there was a considerable increase in endogenous GABA with increasing duration of the NaCl treatment (Fig. 6C). The GABA content in roots increased rapidly, to 5.9-fold its initial level after 3 h of NaCl stress, and to more than 10-fold its initial level at 12 and 24 h of NaCl stress. In the leaves, the GABA content slightly increased after 3 h of NaCl stress, and markedly increased to 4.7–5.6 fold its initial level by 6 h of NaCl stress. These findings suggested that there was a positive correlation between the expression of *CiGAD* genes and GABA accumulation during NaCl stress.

## ABA regulated *CiGAD*s expression under NaCl stress

Treatment of NaCl-stressed seedlings with exogenous ABA significantly affected the expression of both *CiGADs*, and the changes in expression differed between the two genes (Fig. 7). In the roots, both *CiGADs* showed gradually increasing transcript levels with increasing concentrations of exogenous ABA at 3 h. However, *CiGAD1* showed a 3.8-fold

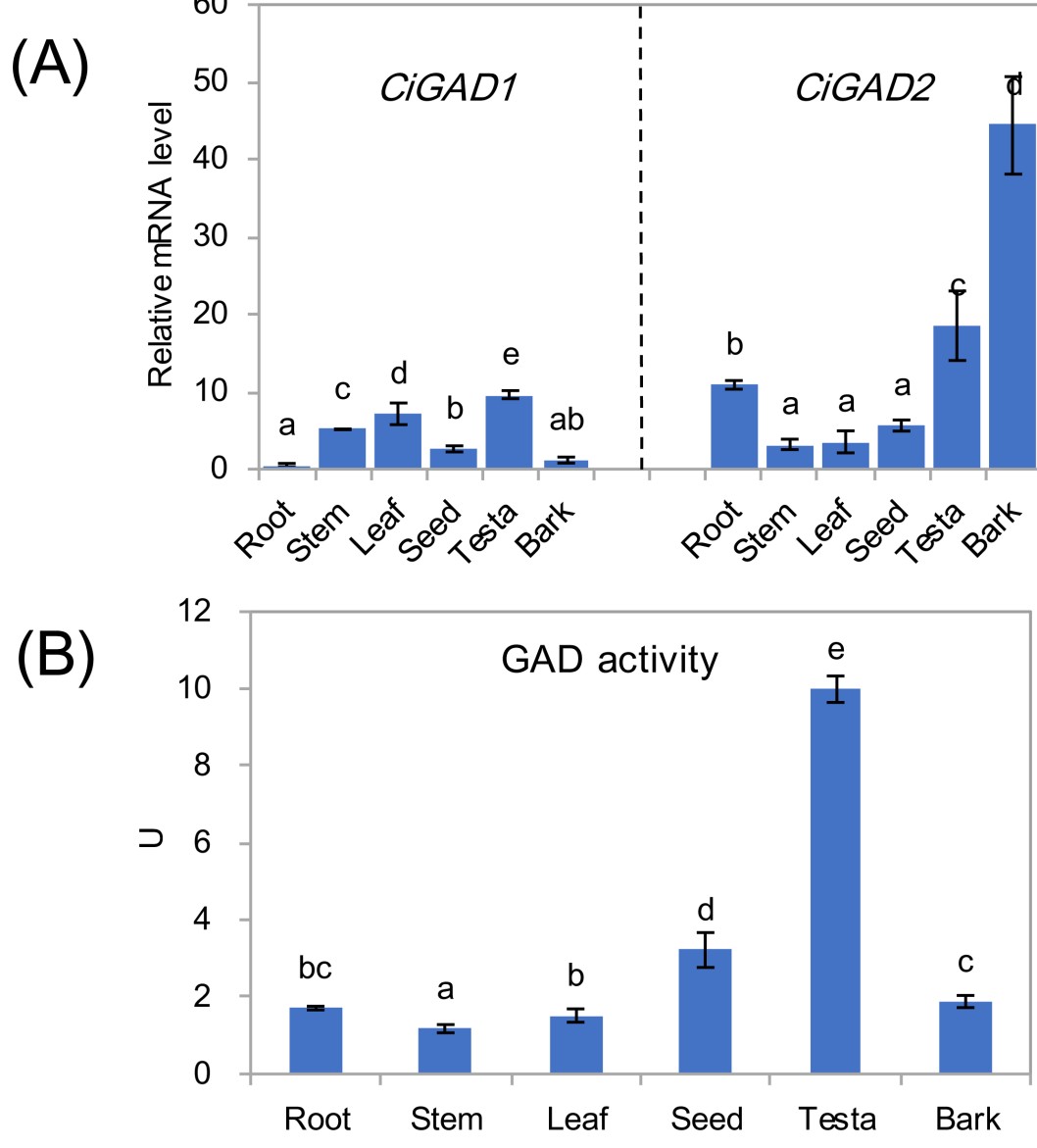

**Figure 4** **Expression of two *CiGADs* (A) and total GAD activity (B) in different tissues of the mature *Caragana intermedia*, including roots, stems, leaves, seeds, testae and bark.** The means and standard errors were calculated from three biological replicates. Different lower-case letters between any two sampling points indicate signification difference at $P < 0.05$ by Duncan's test.

increase in its transcript levels in response to 0.5 μM ABA, similar to the increase in response to 200 μM ABA. The two genes showed almost opposite trends in expression in the roots at 24 h of treatment with ABA; the transcript levels of *CiGAD1* decreased with increasing ABA concentrations, but were still much higher than the transcript levels of *CiGAD2* (Fig. 7A). In the leaves, however, after 3 h of ABA treatment, low concentrations of exogenous ABA (0.5 and 1.0 μM) inhibited the expression of both *CiGADs*, but higher concentrations of ABA induced the expression of both genes. After 24 h of treatment with ABA, 0.5 μM ABA considerably increased the expression levels of both genes to more

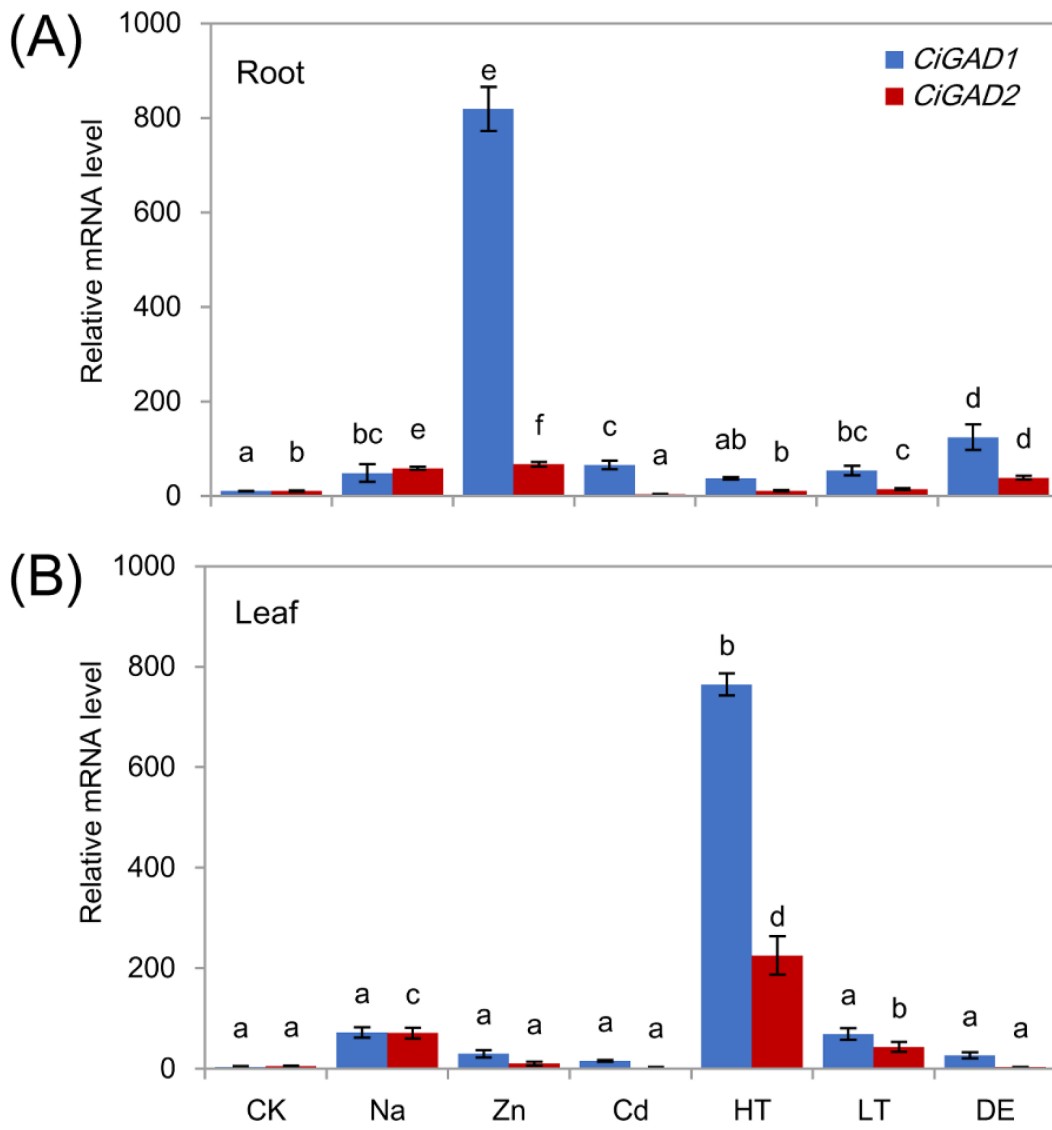

**Figure 5  Stress-specific expression of two *CiGADs* in roots and leaves of young *Caragana intermedia* seedlings under different stress treatments for 3 h.** CK, Water; Na, 200 mM NaCl; Zn, 100 μM ZnSO$_4$; Cd, 500 μM CdCl$_2$; HT, High temperature 40 °C; LT, Low temperature 4 °C; DE, Dehydration. The means and standard errors were calculated from three biological replicates. Different lower-case letters between any two sampling points indicate signification difference at $P < 0.05$ by Duncan's test.

than 5.6-fold that in the control (0 μM ABA), but ABA at other concentrations drastically reduced the transcript levels of both genes, except for *CiGAD1* at 10 μM ABA (Fig. 7B). These results demonstrated that, even at micromolar concentrations, ABA participates in regulating *CiGAD1* and *CiGAD2* transcription during salt stress.

## DISCUSSION

Much attention has been paid to the roles of GABA in the development and stress responses of plants (*Batushansky et al., 2014*; *Batushansky et al., 2015*; *Michaeli & Fromm,*

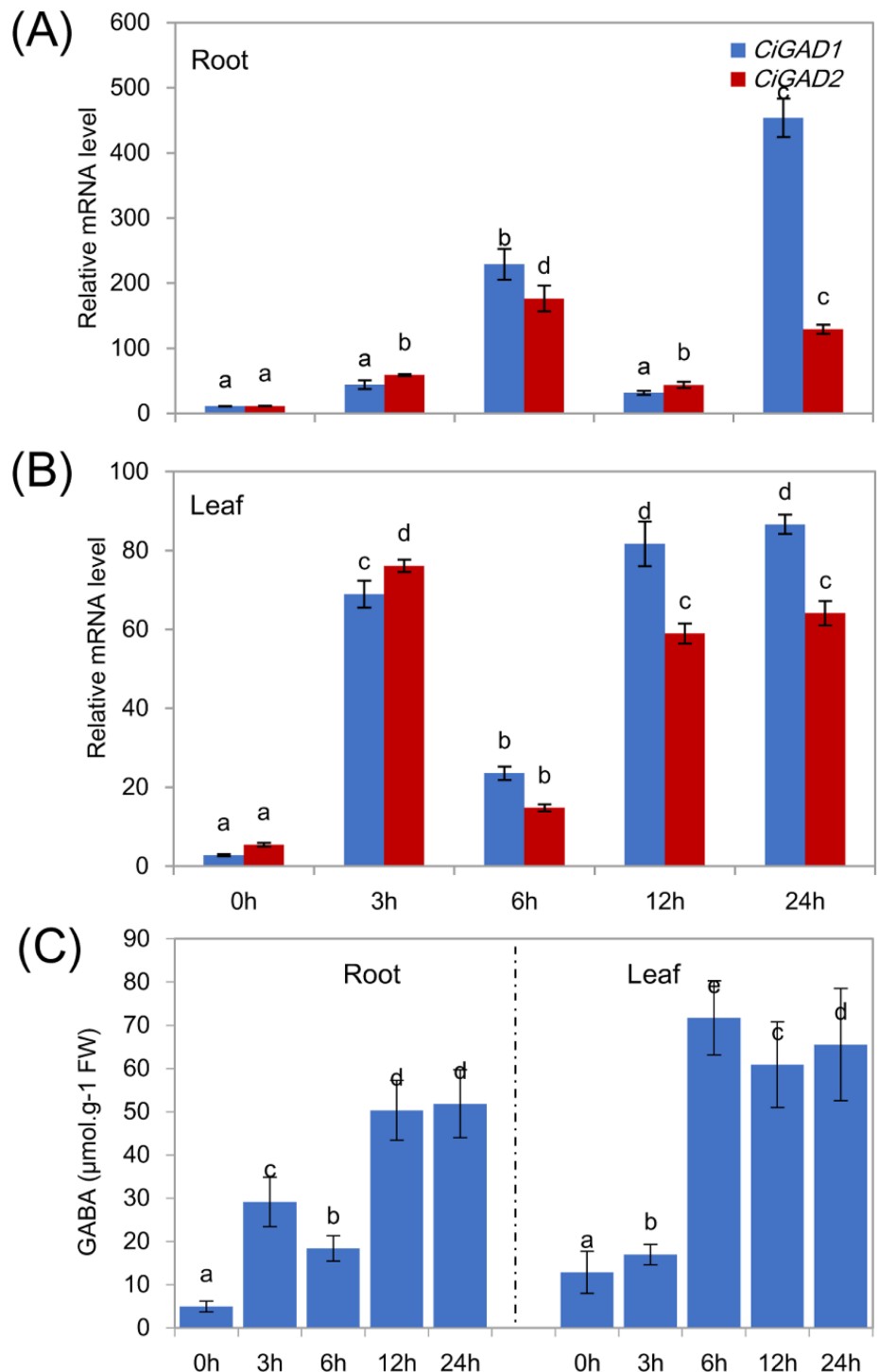

**Figure 6** **Expression of *CiGADs* and production of GABA in roots and leaves of young Caragana inter-media seedlings under 200 mM NaCl treatment for 0, 3, 6, 12 and 24 h.** (A) and (B) *CiGADs*' expression in root and leaves, respectively; (C) GABA production. The means and standard errors were calculated from three biological replicates. Different lower-case letters between any two sampling points indicate sig-nification difference at *P* < 0.05 by Duncan's test.

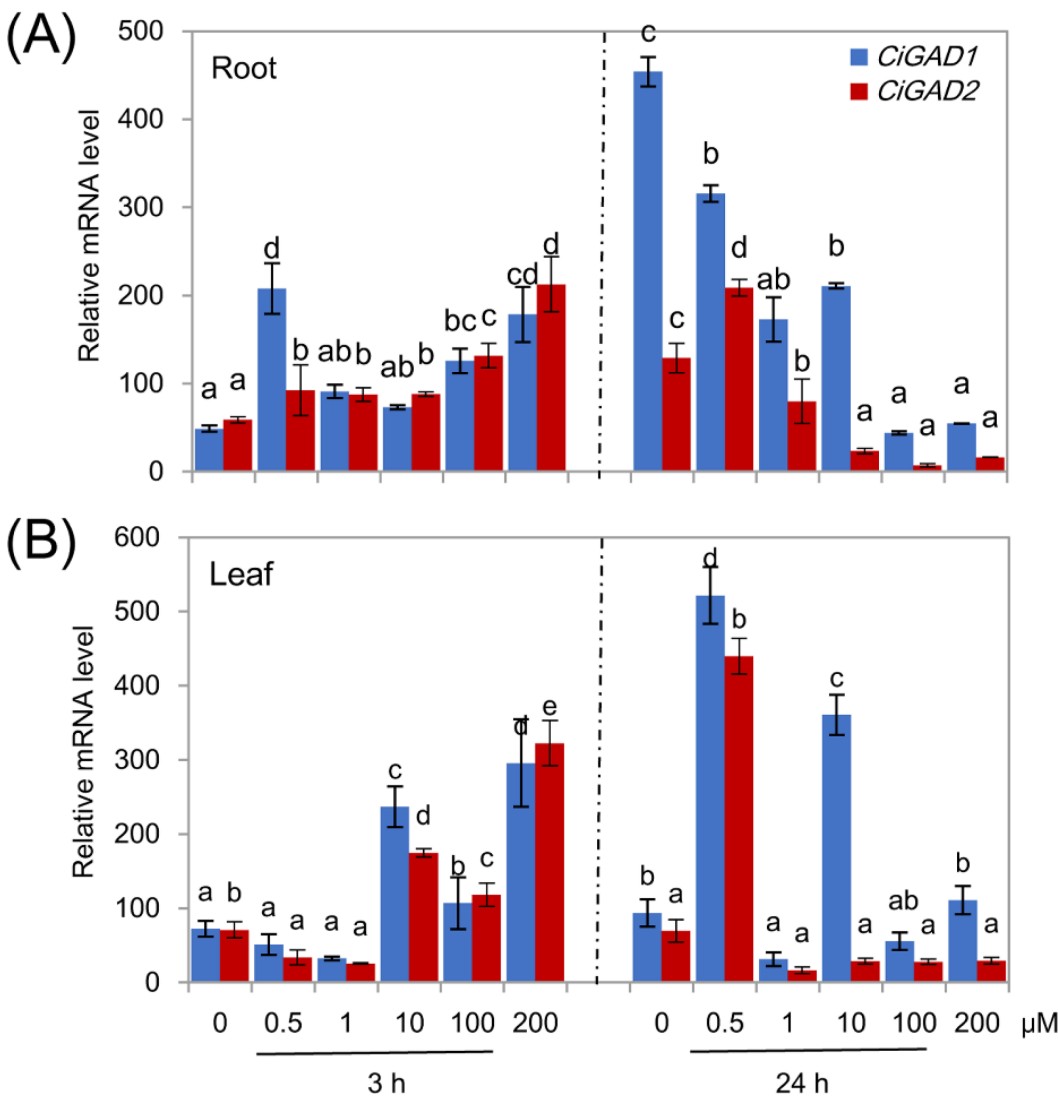

**Figure 7** Effects of ABA on *CiGADs'* expression in roots (A) and leaves (B) of young *Caragana intermedia* seedlings under 200 mM NaCl treatment for 3 and 24 h. The concentration of exogenous ABA was 0, 0.5, 1, 10, 100 and 200 $\mu$M. The means and standard errors were calculated from three biological replicates. Different lower-case letters between any two sampling points indicate signification difference at $P < 0.05$ by Duncan's test.

*2015*; *Molina-Rueda et al., 2015*; *Shi et al., 2010*), and GAD is a key enzyme regulating GABA production and the GABA shunt pathway (*Michaeli & Fromm, 2015*; *Shimajiri et al., 2013*). In this study, two *GAD* genes (*CiGAD1* and *CiGAD2*) were cloned based on specific EST sequences (Supplemental Information 1) identified in our previous work on the molecular response of *C. intermedia* to salt stress. These two genes had ORFs encoding 498- and 494-amino acid residues. Homology analyses showed that these two CiGADs exhibited relatively high similarity (72.1%–88.5%) to GADs from three model plants: soybean, *Arabidopsis*, and poplar, so *CiGADs* were classified into the GAD family (Fig. S2). The CaM/Ca$^{2+}$-dependent GADs conserved the putative CaM-binding domain in plants,

such as *Arabidopsis* (*Turano & Fang, 1998*), tomato (*Gallego et al., 1995*), and *P. ginseng* (*Lee et al., 2010*). But some GADs did not include CaM-binding domain, such as rice GAD2 (*Akama et al., 2001*). Sequence analyses confirmed that both of the CiGADs included a conserved PLP domain and a CaM-binding domain (Fig. 1; *Trobacher et al., 2013*), but whether both CiGADs were indeed activated via CaM or not need a further research (*Mei et al., 2016*). A phylogenetic analysis indicated that *CiGAD1* and *CiGAD2* were much more closely related to *GAD* genes in soybean than to *GAD* genes in the woody plant poplar (Fig. 2). Southern blot analysis confirmed that *CiGAD1* had one copy and *CiGAD2* -related genes were present as two copies in *C. intermedia* (Fig. 3). However, both *AtGAD1* and *AtGAD2* are single copy genes in *Arabidopsis* (*Turano & Fang, 1998*), and a single copy of *GmGAD1* is also observed in soybean (*Matsuyama et al., 2009*). But there are at least two copies of the *PgGAD* in *P. ginseng* (*Lee et al., 2010*). These results indicate that the copy number of *GAD* genes depends on each plant.

The cloning of *CiGAD*s genes will be useful for further research on their functions during plant development and stress responses. As mentioned in the Introduction, *GAD*s are expressed in most tissues, but different *GAD*s show different spatial and temporal expression patterns (*Miyashita & Good, 2008*; *Akama et al., 2001*; *Liu et al., 2014*). Our results showed that *CiGAD1* was mainly expressed in the stems, leaves, and testae; whereas *CiGAD2* was expressed at high levels in bark and was also expressed in the roots and testae (Fig. 4A). The expression patterns indicated that *CiGAD2* might participate in vascular differentiation, like *PpGAD* in maritime pine (*Molina-Rueda et al., 2010*). *Molina-Rueda et al. (2015)* also reported that vascular development was closely linked to GABA production corresponding to *PpGAD* expression. Because the GAD activities were higher in testae and seeds, *CiGAD1* and *CiGAD2* may also play a role in plant reproduction like *GAD*s in citrus fruit (*Liu et al., 2014*) and apples (*Trobacher et al., 2013*).

Several studies have shown that plant *GAD*s expression responds differently to various abiotic stresses in herbaceous plants (*Zhuang et al., 2010*; *Lee et al., 2010*). Our previous findings showed that exogenous GABA can regulate the molecular responses of *C. intermedia* during salt stress via its roles in regulating signaling (reactive oxygen species, ethylene, and ABA-producing enzymes RBOH/ACO/ABA2) and nitrogen metabolism (nitrate transporter, arginase, nodulin) (*Shi et al., 2010*). These results prompted us to analyze the expression patterns of the two identified *CiGAD*s in response to various stresses and exogenous ABA, which orchestrates the interaction between biotic and abiotic stresses via signaling pathways (*Atkinson & Urwin, 2012*). When *Caragana* seedlings were subjected to short-term (3 h) stress treatments including salinity (Na), heavy metals (Zn and Cd), heat (40 °C), cold (4 °C), and drought (3 h dehydration), the transcript levels of *CiGAD1* and *CiGAD2* increased in the roots and leaves (Fig. 3), possibly as a result of an increase in $Ca^{2+}$ induced by each stress treatment (*Akihiro et al., 2008*). The only exception was that *CiGAD2* transcript levels did not increase in response to $Cd^{2+}$ (Figs. 4A, 4B), which is hypothesized to displace metal cofactors (such as $Zn^{2+}$) from proteins or compete with $Ca^{2+}$ to bind to $Ca^{2+}$-binding proteins (*Stohs & Bagchi, 1995*). The salt stress treatment induced *CiGAD1* and *CiGAD2* expression more strongly in leaves (20.6–36.0 fold) than in roots (4.4–5.5 fold) (Figs. 5A, 5B). For comparison, *PgGAD* expression in *P. ginseng*

increased 3-fold after 8 h of salt treatment (*Lee et al., 2010*). This result indicated that the two *CiGAD*s might play a crucial role in the salt stress response in leaves.

In terms of heavy metal stress responses, the Zn treatment significantly increased expression of both *CiGAD*s in leaves and roots, and induced a substantial increase in *CiGAD1* expression (a 74.0-fold increase) in roots (Fig. 5A), whereas Cd had opposite effects on the two genes, inducing an increase in *CiGAD1* expression (4.4–6.0 fold), but no change or a decrease in *CiGAD2* expression (Figs. 5A, 5B). An analysis of the *Arabidopsis* transcriptome showed that *AtGAD4* transcript levels increased 8.4-fold under a 50 $\mu$M Cd$^{2+}$ treatment (*Weber, Trampczynska & Clemens, 2006*). A surprising result in this study was that heat and cold stresses strongly induced expression of *CiGAD1* and *CiGAD2*, especially in leaves; the transcript levels of these genes increased by 218.1-fold and 114.2-fold, respectively, in response to heat, and by 19.7-fold and 21.8-fold, respectively, in response to cold (Fig. 5B). For comparison, *PgGAD* expression in *P. ginseng* increased by 3–5 fold after 8-h heat and cold treatments (*Lee et al., 2010*). Next, we examined changes in the transcript levels of two *CiGAD*s during a 24-h salt treatment, and observed constant increases in their expression (Figs. 6A, 6B) consistent with GABA accumulation (Fig. 6C). The present result suggested that these two *CiGAD*s in *Caragana* were involved in multiple stress responses, especially the responses to Zn and heat stress.

In plants, the phytohormone ABA is a key endogenous signaling molecule in the responses to various stresses. Understanding ABA-signaling is essential for improving plant performance in the future (*Atkinson & Urwin, 2012*; *Raghavendra et al., 2010*). In our previous study, we found that exogenous GABA induced the expression of the ABA-biosynthetic gene *ABA2* in *Caragana* under salt stress (*Shi et al., 2010*). Exogenous ABA has also been shown to enhance GAD activity and increase the GABA content in fava bean (*Yang, Hui & Gu, 2016*), and to induce the expression of *ZmGAD1*, which harbors ABA-related *cis*-elements in its promoter (*Zhuang et al., 2010*). In the present study, we found that exogenous ABA increased the transcript levels of *CiGAD1* and *CiGAD2* at an early stage (3 h) of the salt treatment (Figs. 7A, 7B); while after 24 h of salt treatment, the transcript levels of both genes were reduced in the roots of ABA-treated plants, but strongly increased in the leaves of plants treated with 0.5 $\mu$M ABA (Figs. 7A, 7B). These different transcriptional responses of *CiGAD1* and *CiGAD2* to exogenous ABA may be because of differences in ABREs in their promoter regions (*Liu et al., 2014*).

In conclusion, the results of this study showed that two *CiGAD*s cloned from *Caragana* are closely related to homologs in another legume, soybean. The transcript levels of *CiGAD2* were much higher than those of *CiGAD1* in bark, suggesting that *CiGAD2* might play a role in secondary growth of woody plants. The GAD activities were higher in testae and seeds than other tissues, *CiGAD*s may also play a role in plant reproduction. Multiple stresses significantly affected the transcript levels of both *CiGAD*s. Interestingly, Zn and heat stresses had the strongest effects on *CiGAD1* expression, indicating that *CiGAD1* plays important roles in the responses to Zn and heat stresses. Our results also suggested that ABA is involved in regulating these two *CiGAD*s during stress in *Caragana*, an important nitrogen-fixing legume shrub. These results will be useful for further research on the role of

*GADs* in carbon and nitrogen metabolism and in signal transduction during the responses to environmental stresses.

### Funding

This work was supported by the National Natural Science Foundation of China (31100490), the Special Fund for Nonprofit Research Institute in Central Government (CAFYBB2012040; CAFYBB2014ZX001-3), and the Lecture and Study Program for Outstanding Scholars from Home and Abroad (CAFYBB2011007). The funders had no role in study design, data collection and analysis, decision to publish, or preparation of the manuscript.

### Grant Disclosures

The following grant information was disclosed by the authors:
National Natural Science Foundation of China: 31100490.
Nonprofit Research Institute in Central Government: CAFYBB2012040, CAFYBB2014ZX001-3.
Lecture and Study Program: CAFYBB2011007.

### Competing Interests

The authors declare there are no competing interests.

### Author Contributions

- Jing Ji conceived and designed the experiments, performed the experiments, analyzed the data, wrote the paper, prepared figures and/or tables, reviewed drafts of the paper.
- Lingyu Zheng conceived and designed the experiments, performed the experiments, analyzed the data, reviewed drafts of the paper.
- Jianyun Yue, Xiamei Yao, Ermei Chang, Tiantian Xie, Nan Deng and Yuwen Huang performed the experiments, reviewed drafts of the paper.
- Lanzhen Chen analyzed the data, contributed reagents/materials/analysis tools, reviewed drafts of the paper.
- Zeping Jiang conceived and designed the experiments, contributed reagents/materials/-analysis tools, reviewed drafts of the paper.
- Shengqing Shi conceived and designed the experiments, analyzed the data, contributed reagents/materials/analysis tools, wrote the paper, prepared figures and/or tables, reviewed drafts of the paper.

### DNA Deposition

The following information was supplied regarding the deposition of DNA sequences:
CiGAD1 and CiGAD2 have been deposited in GenBank under the accession numbers KU586714 and KU586715, respectively.

## Data Availability

The raw data is included in Figs. 1 through 6 in the manuscript, and in the Supplementary Files.

## Supplemental Information

Supplemental information for this article can be found online at http://dx.doi.org/10.7717/peerj.3439#supplemental-information.

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
