# Peer review of "Identification of two CiGADs from Caragana intermedia and their transcriptional responses to abiotic stresses and exogenous abscisic acid"

_PeerJ, doi:10.7717/peerj.3439_

## Round 0.1 · original submission · Major Revisions

After careful evaluation of your manuscript by the two expert reviewer and by myself I want to give you the following recommendation:

One critical point in your manuscript in my opinion and in agreement with Reviewer 1 is the fact that the complete number of GAD isoforms is not analyzed. Therefore the performance of a Southern analysis is recommended to get more information about the exact number of genes present in the C. intermedia genome, what is very important to evaluate the role of both GAD isoforms in abiotic stress response.

Another very important issue is that the activation of GAD isoforms via CaM is not discussed in due detail. It is recommended to present more information and discussion whether both GAD isoforms are indeed activated via CaM or not.

A third recommendation for the preparation of the revised manuscript is the re-writing of some parts of the experimental procedures (number of experiments and replicates) and the discussion (GABA, NaCl treatment) since in its present form these paragraphs are not very clear.

Reviewer 1 ·

Basic reporting

I have no comments.

Experimental design

I have no comments.

Validity of the findings

I have no comments.

Additional comments

Comments to the Authors:
The Authors identified two different cDNA clones encoding glutamate decarboxylase (GAD) from Caragana intermedia and analyzed their transcription level in response to abiotic stresses and ABA. The MS looks interesting, especially several abiotic stresses strongly induced CiGAD transcription. However, I feel that several data are missing for publication as follows:
First, authors identified two GAD genes in this study, the possibility remains that more genes are present in the genome. At least Southern analysis should be required to know the copy number in the genome. Second, the Authors addressed CiGAD1 and CiGAD2 are both carrying a CaM-binding domain at the C-terminus. But recent study by Mei et al (2016) published in Scientific Reports reported that stress-induced tea GAD gene (CsGAD2) does not has an ability to bind to a CaM, while CsGAD1 that can bind to a CaM is likely to activate via CaM in response to abiotic stress. At least from alignment, they seemed to carry them. Because both CiGAD genes transcriptionally activate in response to abiotic stresses, it would be very important whether these GAD isoforms indeed are activated via CaM or not. At least CaM-bind assay should be needed in this study,using recombinant CiGAD proteins. Also an in vitro enzymatic analysis would be recommended in the absence or presence of Ca2+/CaM.

·

Basic reporting

No comments.

Experimental design

It was not clear whether 3 different experimental series were conducted or only from 1 experimental series 3 biological replicates were taken. This should be clarified.

Validity of the findings

No comments.

Additional comments

In Materials and Methods section of the manuscript plant materials and treatments should be re-written in its present form it was not clear.
GABA determination method should be written in detail, it was written very briefly.
It was not clear whether 3 different experimental series were conducted or only from 1 experimental series 3 biological replicates were taken. This should be clarified.
Discussion should be re-written since in its present form rather than discussing the results it looks like a review of publications about GABA.
GAD transcript levels were highest of all in Zn and heat stress in root and leaf respectively. Therefore the authors should explain why they chose to use NaCl treated groups for further analysis.
In order to provide a better comparison between GAD activities of different tissues/organs GAD activity levels should be expressed as specific activity (unit mg protein-1).

---

## Round 0.2 · Minor Revisions

Your manuscript has been revised according to the points mentioned previously, however, there are still a few minor points addressed by reviewer 1 that needs further clarification.

Reviewer 1 ·

Basic reporting

no comment

Experimental design

See general comment below.

Validity of the findings

no comment

Additional comments

The response to the Authors’ comments 1, 2 and revised manuscript

1 The Authors performed Southern blot analysis to estimate copy number of the GAD gene in C. intermedia genome, showing one copy of the CiGAD1gene and two copies of the CiGAD2 gene. Although an introduction of Southern blot data is highly evaluated, several issues should be clarified:

1.1 Line 31: … and CiGAD2 (but not CiGAD1) had two copies in C. intermedia.
1.2 Line 145: indicate amount (ug) of genome DNA used.
1.3 Lines 147-148: indicate probe regions of CiGAD1 and CiGAD2 cDNA. Used 3’-UTR specific region of them or full-length cDNA?
1.4 Lines 204-212: it would be better to show identity (%) between CiGAD1 and CiGAD2 in the nucleotide level.
1.5 Lines 228-229: from Southern blot analysis, it is clear that CiGAD1 is a single copy and CiGAD2-related gene are present as two copies but not two copies of CiGAD2, because DNA probe of CiGAD2 resulted just in detection of two related DNA sequence. You cannot conclude that they are identical, most probably another sequence detected might be paralog of CiGAD2. It is the similar relation of GmGAD2 and GmGAD4 in Figure 2. Therefore, you can change several sentences concerning copy number in Abstract, results and Discussion.
1.6 Figure 3: Molecular DNA marker (M) is a range from 250 bp to 10, 000 bp, not protein maker (250 to 10 kDa). What is blank and a strong signal around 750?

2 The authors explained isolation of the CiGAD1 and/or CiGAD2 fusion proteins from yeast for in vitro assay of CaM-binding ability and enzymatic analysis, but neither of them worked. Previous works were successfully done using E. coli system. Is there any special reason to attempt in yeast?

·

Basic reporting

The manuscript has been revised according to the points mentioned in my previous review.

Experimental design

The manuscript has been revised according to the points mentioned in my previous review.

Validity of the findings

The manuscript has been revised according to the points mentioned in my previous review.

Additional comments

The manuscript has been revised according to the points mentioned in my previous review.

---

## Round 0.3 · accepted · Accept

I am happy to inform you that your manuscript is now accepted and I would like to congratulate on this good publication.